# Progressive Ensemble Distillation: Building Ensembles for Efficient Inference

**Don Kurian Dennis**
Carnegie Mellon University

**Abhishek Shetty**
University of California, Berkeley

**Anish Sevekari**
Carnegie Mellon University

**Kazuhito Koishida**
Microsoft

**Virginia Smith**
Carnegie Mellon University

## Abstract

We study the problem of *progressive ensemble distillation*: Given a large, pretrained teacher model $g$, we seek to decompose the model into smaller, low-inference cost student models $f_i$, such that progressively evaluating additional models in this ensemble leads to improved predictions. The resulting ensemble allows for flexibly tuning accuracy vs. inference cost at runtime, which is useful for a number of applications in on-device inference. The method we propose, B-DISTIL, relies on an algorithmic procedure that uses function composition over intermediate activations to construct expressive ensembles with similar performance as $g$, but with smaller student models. We demonstrate the effectiveness of B-DISTIL by decomposing pretrained models across standard image, speech, and sensor datasets. We also provide theoretical guarantees in terms of convergence and generalization.

## 1 Introduction

Knowledge distillation aims to transfer the knowledge of a large model into a smaller one [5, 23]. While this technique is commonly used for model compression, one downside is that the procedure is fairly rigid—resulting in a single compressed model of a fixed size. In this work, we instead consider the problem of *progressive ensemble distillation*: approximating a large model via an ensemble of smaller, low-latency models such that such that progressively evaluating additional models in this ensemble leads to improved predictions. The resulting decomposition is useful for many applications in on-device and low-latency inference. For example, components of the ensemble can be selectively combined to flexibly meet accuracy/latency constraints [31, 44], can enable efficient parallel inference execution schemes, and can facilitate *early-exit* [4, 11] or *anytime inference* [36, 28] applications, which are scenarios where inference may be interrupted due to variable resource availability.

More specifically, our work seeks to distill a large pretrained model, $g$, onto an ensemble of 'smaller' models, such that evaluating the first model produces a coarse estimate of the prediction (e.g., covering common cases), and evaluating additional models improves on this estimate (see Figure 1). There are major advantages to such an ensemble for on-device efficient inference.

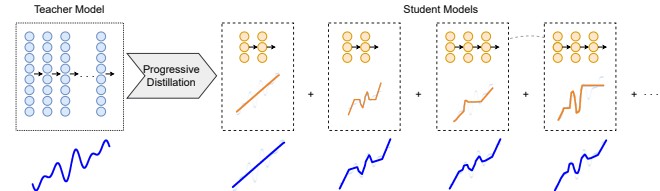

**Figure 1:** In progressive ensemble distillation, a large teacher model is distilled into an ensemble of low inference cost models. The more student models we evaluate, the closer the ensemble's decision boundary is to that of the teacher model. Models in the ensemble are allowed to depend on previously computed features to reduce overhead and inference cost.

Corresponding author: Don Dennis <dondennis@cmu.edu>.

37th Conference on Neural Information Processing Systems (NeurIPS 2023).

Concretely, (i) inference cost vs. accuracy trade-offs can be controlled on-demand at execution time, (ii) the ensemble can either be executed in parallel or in sequence, or possibly a mix of both, and (iii) we can improve upon coarse initial predictions without re-evaluation at runtime.

While traditional distillation methods are effective when transferring information to a single model of similar capacity, it has been shown that performance can degrade significantly when reducing the capacity of the student model [34, 19]. Moreover, distillation of a deep network onto a weighted sum of shallow networks rarely performs better than distillation onto a single model [9, 1].

Our insight in this work is that by composing and reusing activations and by explicitly incentivizing models to be weak learners during distillation, we can successfully find weak learners even when the capacity gap is relatively large. As long as these composition functions are resource-efficient, we are able to increase our hypothesis class capacity at roughly the same inference cost as a single model. Moreover, we show that our procedure retains the theoretical guarantees of classical boosting methods [39]. Concretely, we make the following contributions:

- We formulate progressive ensemble distillation as a two player zero-sum game, derive a weak learning condition for distillation, and present our algorithm, B-DISTIL, to approximately solve this game. To make the search for weak learners in low parameter count models feasible, we solve a log-barrier based relaxation of our weak learning condition. By allowing models to reuse computation from select intermediate layers of previously evaluated models of the ensemble, we can increase the model's capacity without significantly increasing inference cost.

- We empirically evaluate our algorithm on synthetic and real-world classification tasks from computer vision, speech, and sensor processing with models suitable for the respective domains. We show that our ensemble behaves like a decomposition, allowing a run-time trade-off between accuracy and computation, while retaining competitive performance with the teacher model.

- We provide theoretical guarantees for our algorithm in terms of in-sample convergence and generalization performance. Our framework is not architecture or task specific and can recover existing ensemble models used in efficient inference; we believe our work thus puts forth a general lens to view previous work and also to develop new, principled approaches for efficient inference.

## 2   Background and Related Work

**Efficient Inference.**   Machine learning inference is often resource-constrained when deployed in practical applications due to memory, energy, cost, or latency constraints. This has spurred the development of numerous techniques for efficient inference. Pruning and approximations of pre-trained parameter tensors through low-rank, sparse and quantized representations [22, 3, 18, 20] have been effective is reducing resource requirements. There are also architecture and task specific techniques for efficient inference [12, 45, 13]. In contrast to compressing an already trained model, algorithms have also been developed to train compressed models by incorporating resource constraints as part of their training routines [2, 7]. More recently, algorithms have been developed to search and find smaller sub-models from a single pre-trained model [46, 6].

**Knowledge distillation.**   Knowledge distillation aims to transfer the knowledge of a larger model (or model ensemble) to a smaller one [5, 23]. Despite its popularity, performing compression via distillation has several known pitfalls. Most notably, it is well-documented that distillation performs poorly when there is a *capacity gap*, i.e., the teacher is significantly larger than the student [34, 19, 9, 1]. When performing distillation onto a weighted combination of ensembles, it has been observed that adding additional models into the ensemble does not dramatically improve performance over that of a single distilled model [1]. There is also a lack of theoretical work characterizing when and why distillation is effective for compression [21]. Our work aims to address these pitfalls by developing a principled approach for progressively distilling a large model onto an ensemble of smaller, low-capacity ones. We defer readers to [21] for a recent survey on varied applications of and approaches for distillation at large.

**Early exits and anytime inference.**   Many applications stand to benefit from the output of progressive ensemble distillation, which allows for flexibly tuning accuracy vs. inference cost and executing inference in parallel. Enabling trade-offs between accuracy and inference cost is particularly useful for applications that use early exit or anytime inference schemes. In on-device continuous (online) inference settings, *early exit* models aim to evaluate common cases quickly in order to improve

energy efficiency and prolong battery life [11, 4]. For instance, a battery powered device continuously listening for voice commands can use early exit methods to improve battery efficiency by quickly classifying non-command speech. Many early exit methods are also applicable to anytime inference [28, 36]. In *anytime inference*, the aim is to produce a prediction even when inference is interrupted, e.g., due to resource contention or a scheduler decision. Unlike early exit methods where the classifier chooses when to exit, anytime inference methods have no control over when they are interrupted. We explore the effectiveness of our method, B-DISTIL, for such applications in Section 5.

**Two-player games, online optimization and boosting.** In this work we formulate progressive ensemble distillation as a two player zero-sum game. The importance of equilibrium of two player zero-sum games have been recognized since the foundational work of von Neumann and Morgenstern [42]. Later applications by Schapire [38] and Freund [15], Freund and Schapire [16] identified close connections between boosting and two-player games. On the basis of this result, a number of practical boosting-based learning approaches such as AdaBoost [17], gradient boosting [33], and XGboost [8] have been developed. Boosting has only recently seen success in modern deep learning applications. In particular, Suggala et al. [40] propose a generalized boosting framework to *train* boosting based ensembles of deep networks. Their key insight is that allowing function compositions in feature space can help boost deep neural networks. Although they focus on training and do not produce decompositions of pretrained models, we use a similar approach in our work to select intermediate layers connections between ensemble components (Section 3.3). A more general application of boosting that is similar to our setup is by Trevisan et al. [41]. They prove that given a target bounded function $g$ (e.g., the teacher model) and class of candidate approximating functions $f \in \mathcal{F}$, one can iteratively approximate $g$ arbitrarily well with respect to $\mathcal{F}$ using ideas from online learning and boosting. However, these results depend on the ability to find a function $f_t$ in iteration $t$ that leads to at least a small constant improvement in a round-dependent approximation loss. A key contribution of our work is showing that such functions can be found for the practical application of progressive ensemble distillation by carefully selecting candidate models.

## 3 Progressive Ensemble Distillation with B-DISTIL

As discussed, our goal in performing progressive ensemble distillation is to approximate a large model via an ensemble of smaller, low-latency models so that we can easily trade-off between accuracy and inference-time/latency at runtime. In this section we formalize the problem of progressive ensemble distillation as a two player game and discuss our proposed algorithm, B-DISTIL.

### 3.1 Problem Formulation

Consider repeated plays of a general two player zero-sum game with the pure strategy sets comprising of a hypothesis class $\mathcal{F}$ and a probability distribution $\mathcal{P}$. Given a loss function $\mathcal{L}$, we let the loss (and reward) of the players be given by $F(f, p) = \mathbb{E}_{x \sim p}[\mathcal{L}(f, x)]$, and the minimax value of the game is:

$$\max_{p \in \mathcal{P}} \min_{f \in \mathcal{F}} F(f, p). \tag{1}$$

In the context of distillation, given a training set $\{x_i\}$, we can think of the role of the max player in Equation (1) as producing distributions over the training set and the role of the min player as producing a hypothesis that minimizes the loss on this distribution. In this setting, note that $\mathcal{P} = \left\{ p \in \mathbb{R}^{N \times L} : p_{i,j} \geq 0 \quad \forall j \sum_i p_{i,j} = 1 \right\}$ is the product of simplices in $N \times L$ dimensions, and

$$(\nabla_f F(f, p))_j = \sum_i p_{i,j} (f(x_i) - g(x_i))_j. \tag{2}$$

Our goal is to produce an ensemble of predictors from the set of hypothesis classes $\{\mathcal{F}_m\}$ to approximate $g$ 'well'. We now appeal to tools from the framework of stochastic minimax optimization to approximately attain the value in Equation (1) (see Appendix A for a more involved discussion). As is common in this setup, we assume our algorithm is provided access *weak gradient* vector $h$ such that, when queried at distribution $p \in \mathcal{P}$ and for $\beta > 0$,

$$\langle h, \nabla_f F(f, p) \rangle \geq \beta. \tag{3}$$

We perform this construction iteratively by searching for weak learners or weak gradients in the sense of Equation (3) with respect to the target $g$ in the class $\mathcal{F}_m$. Conditioned on a successful search we can guarantee in-sample convergence to the minimax value in Equation (1) (Theorem 1) and bound the excess risk of the ensemble (Theorem 2). Although Equation (3) is a seemingly easier notion than the full optimization, in many problems of interest even this is challenging. In fact, in the multilabel setting that we focus on, one of the main algorithmic challenges is to construct an algorithm that can reliably find low cost weak gradients/learners (see Section 3.3).

**Algorithm 1** B-DISTIL: Main algorithm

**Require:** Target $g$, rounds $T$, data $\{(x_i, y_i)\}_{i=1}^N$, learning rate $\eta$, model classes $\{\mathcal{F}_m\}_{m=1}^M$
1: $K_t^+(i,j), K_t^-(i,j) \leftarrow \frac{1}{2N}, \frac{1}{2N} \quad \forall (i,j)$
2: $F, r, t \leftarrow \emptyset, 1, 1$
3: **while** $r < R$ and $t < T$ **do**
4: $\quad f_t = \text{FIND-WL}(K_t^+, K_t^-, \mathcal{F}_r)$
5: $\quad$ **if** $f_t$ is NONE **then**
6: $\quad\quad r \leftarrow r + 1$
7: $\quad\quad$ continue
8: $\quad$ **end if**
9: $\quad$ With $l := f_t - g$, update $K_t^+, K_t^-$. $\forall (i,j)$

$$K_{t+1}^+(i,j) \leftarrow K_t^+(i,j) \exp(-\eta \cdot l(x_i)_j) \quad (4)$$

$$K_{t+1}^-(i,j) \leftarrow K_t^-(i,j) \exp(\eta \cdot l(x_i)_j) \quad (5)$$

10: $\quad$ Normalize $K_t^+, K_t^-$.
11: $\quad F, t \leftarrow F \cup \{f_t\}, t + 1$
12: **end while**
13: **Return** $\frac{1}{|F|} \sum_{i=1}^{|F|} f_i$

---

**Algorithm 2** FIND-WL

**Require:** Probability matrices $K^+, K^-$, model class $\mathcal{F}$ parameterized by $\theta \in \Theta$, hyperparameters for SGD

1: Obtain $\{\mathcal{F}_r\}_1^R$ by expanding $\mathcal{F}$ (Section 3.2).
2: **for** $\mathcal{F}' \in \{\mathcal{F}_r\}_{r=1}^R$ **do**
3: $\quad$ Initialize initial parameter $\theta_0 \in \mathcal{F}'$.
4: $\quad$ **for** $i \in \{1, \ldots, \text{max-search}\}$ **do**
5: $\quad\quad$ Randomly initialize $f_{\theta_i}$.
6: $\quad\quad$ Run SGD to solve Equation (6).
7: $\quad\quad$ **if** $f_{\theta_i}$ is a weak learner **then**
8: $\quad\quad\quad$ **Return** $f_{\theta_i}$
9: $\quad\quad$ **end if**
10: $\quad$ **end for**
11: **end for**
12: **Return** NONE

## 3.2 B-DISTIL Algorithm

Concretely, at each round $t$, our proposed algorithm, B-DISTIL, maintains matrices $K_t^+ \in R^{N \times L}$ and $K_t^- \in R^{N \times L}$ of probabilities (in our setting, it turns out to be easier to maintain the positive errors and the negative error separately). Note that the matrices $K_t^+$ and $K_t^-$ are such that for all $j \in [L]$, $\sum_i K_t^+(i,j) + K_t^-(i,j) = 1$. Moreover, for all $(i,j) \in [N] \times [L]$, $0 \leq K_t^+(i,j), K_t^-(i,j) \leq 1$. The elements $K_t^+(i,j)$ and $K_t^-(i,j)$ can be thought of as the weight on the residual errors $f_{t-1}(x) - g(x)$ and $g(x) - f_{t-1}(x)$ respectively, up-weighting large deviations from the teacher model $g(x)$. We formalize our notion of weak learners in this setting using Definition 1, which can be seen as a natural extension of the standard weak learning assumption in the boosting literature.

**Definition 1** (Weak learning condition). *Given a dataset $\{(x_i, y_i)\}_{i=1}^N$, a target function $g : \mathcal{X} \to \mathbb{R}^L$ and probability matrices $K_t^+, K_t^-$, a function $f_t : \mathcal{X} \to \mathbb{R}^L$ is said to satisfy the weak learning condition with respect to $g$, $\forall j$, if the following sum is strictly positive:*

$$\sum_i K_t^+(i,j)(f_t(x_i) - g(x_i))_j + K_t^-(i,j)(g(x_i) - f_t(x_i))_j.$$

At each round $t$, with the current probability matrices $K_t^-, K_t^+$, B-DISTIL performs two steps; first, it invokes a subroutine FIND-WL (discussed below) that attempts to find a classifier $f_t \in \mathcal{F}_r$ satisfying the weak learning condition (Definition 1). If such a predictor is found, we add it to our ensemble and proceed to the second step, updating the probability matrices $K_t^-, K_t^+$ based on errors made by $f_t$. This is similar in spirit to boosting algorithms such as AdaBoost [39] for binary classification. If no such predictor can be found, we invoke the subroutine with the next class, $\mathcal{F}_{r+1}$, and repeat the search till a weak learner is found or we have no more classes to search in.

## 3.3 Finding Weak Learners

As mentioned above, the main difficulty in provably approximating the teacher model in this setting is in finding a *single* learner $f_t$ at round $t$ that satisfies our weak learning condition simultaneously for all labels $j$. Existing boosting methods for classification treat multi-class settings ($L > 1$) as $L$ instances of the binary classification problem (one vs. all) [39]. They typically choose $L$ different weak learners for each instance, which is unsuitable for resource efficient on-device inference. The difficulty is further increased by the capacity gap between the student and teacher models we consider for distillation. Thus, along with controlling temperature for distillation, we employ two additional strategies: 1) we use a regularizer in the objective FIND-WL solves to promote weak learning and, 2) we efficiently reuse a limited number of stored activation outputs of previously evaluated models to increase the expressivity of the current base class.

**Log-barrier regularizer.** To find a weak learner, the FIND-WL method minimizes the sum of two loss terms using stochastic gradient descent. The first is standard binary/multi-class cross-entropy distillation loss [23], with temperature smoothing. The second term is defined in Equation (6):

$$-\frac{1}{\gamma}\sum_{i,j} I_{ij}^{+} \log\left(1 + \frac{l(x_i)_j}{2B}\right) + (1 - I_{ij}^{+})\log\left(1 - \frac{l(x_i)_j}{2B}\right) \tag{6}$$

Here $I_{ij}^{+} := I[K_t^{+}(i,j) > K_t^{-}(i,j)]$, $B$ is an upper bound on the magnitude of the logits, and $l(x_i) := f(x_i) - g(x_i)$. To see the intuition behind Equation (6), assume the following holds; $\forall (i,j)$,

$$(K_t^{+}(i,j) - K_t^{-}(i,j))(f(x_i) - g(x_i))_j > 0. \tag{7}$$

Summing over all $x_i$, we can see that this is sufficient for $f$ to be a weak learner with respect to $g$. Equation (6) is a soft log-barrier version of the weak learning condition, that penalizes those $(i,j)$ for which Equation (7) does not hold. By tuning $\gamma$ we can increase the relative importance of the regularization objective, encouraging $f_t$ to be a weak learner potentially at the expense of classification performance.

**Intermediate layer connections and profiling.** As discussed in Section 2, distillation onto a linear combination of low capacity student models often offers no better performance than that of any single model in the ensemble trained independently. For boosting, empirically we see that once the first weak learner has been found in some class $\mathcal{F}_m$ of low-capacity deep networks, it is difficult to find a weak learner for the reweighed objective from the same class $\mathcal{F}_m$. To work around this we let our class of weak learners at round $t$ include functions that depend on the output of intermediate layers of previous weak learners [40].

As a concrete example, consider a deep fully connected network with $U$ layers, parameterized as $f = W\phi_{1:U}$. Here $\phi_{1:u}$ can be thought of as a feature transform on $x$ using the first $u$ layers into $\mathbb{R}^{d_u}$ and $W \in R^{d_U \times L}$ is a linear transform. With two layer fully connected base model class $\mathcal{F}_0 := \{W^{(0)}\phi_{1:2}^{(0)} \mid W^{(0)} \in \mathbb{R}^{L \times d_2}\}$ (dropping subscript $m$ for simplicity), we define:

$$\mathcal{F}_r = \{W^{(r)}\phi_{1:2}^{(r)}(\text{id} + \phi_{1:2}^{(r-1)})\} \quad \text{and,} \quad \mathcal{F}'_r\{W^{(r)}\phi_2^{(r)}(\text{id} + \phi_1^{(r)} + \phi_1^{(r-1)})\},$$

with $\text{id}(x) := x$. It can be seen that $\{\mathcal{F}_r\}$ and $\{\mathcal{F}'_r\}$ define a variant of residual connection based networks [27]. It can be shown that classes of function $\{\mathcal{F}_r\}$ (and $\{\mathcal{F}'_r\}$) increase in capacity with $r$. Moreover, when evaluating sequentially the inference cost of a model from $\mathcal{F}_r$ is roughly equal to that of $\mathcal{F}$, since each subsequent evaluation *reuses* stored activations from previous evaluations. For this reason the parameter count of each $\mathcal{F}_r$ remains the same as that of the base class. Note that by picking the base class as dense networks at various scales and connections as dense connections our algorithm can recover MSDNets studied in [27]. Similarly, by picking the base class as root nodes, and connections as binary connections, we recover an HNE from [36].

We informally refer to the process of constructing $\{\mathcal{F}_r\}$ given a choice of base class $\mathcal{F}_0$, the parameter $R$ and the connection type as *expanding* $\mathcal{F}_0$. Note that while intermediate connections help with capacity, they often reduce parallelizability as models become mutually dependent. As a practical consequence dependencies on activation outputs of later layers are preferred, and we use the Rasley et al. [35] profiler to measure inference cost during training rounds and rank models (see Appendix C).

## 4 Theoretical Analysis

In this section we provide theoretical analysis and justification for our method. First, we show that the ensemble output produced by algorithm 1 converges to $g$ at $\mathcal{O}(1/\sqrt{T})$ rate, provided that the procedure FIND-WL succeeds at every time $t$.

**Theorem 1.** *Suppose the class $\mathcal{F}$ satisfies that for all $f \in \mathcal{F}$, $\|f - g\|_\infty \leq G_\infty$. Let $F = \{f_t\}$ be the ensemble after $T$ rounds of Algorithm 1, with the final output $F_t = \frac{1}{T}\sum_{t=1}^{T} f_t$. If $f_t$ satisfies eq. (7) for all $t \leq T$ then for $T \geq \ln 2N$ and $\eta = \frac{1}{G_\infty}\sqrt{\frac{\ln 2N}{T}}$, we have for all $j$*

$$\|F_{t,j} - g_j\|_\infty \leq G_\infty \sqrt{\frac{\ln 2N}{T}}, \tag{8}$$

*where $F_{t,j}$ and $g_j$ are the $j^{th}$ coordinates of the functions $F_t$ and $g$ respectively.*

We defer the details of the proof to the Appendix B. The main idea behind the proof is to bound the rate of convergence of the algorithm towards the minimax solution. This proceeds by maintaining a potential function and keeping track of its progress through the algorithm. The bounds and techniques here are general in the sense that for various objectives and loss functions appropriate designed weak learners give similar rates of convergence to the minimax solution. Furthermore, a stronger version that shows exponential rates can be shown by additionally assuming an edge for the weak learner.

In addition to the claim above about the in-sample risk, we also show that the algorithm has a strong out-of-sample guarantee. We show this by bounding the generalization error of the algorithm in terms of the generalization error of the class $\mathcal{F}$. In the following theorem, we restrict to the case of binary classification for simplicity, but the general result follow along similar lines. Let $\mathcal{C}_T$ denote the class of functions of the form

$$F_T(x) = \text{sign}(\tfrac{1}{T} \sum_{i=1}^{T} f_t(x)),$$

where $f_t$ are functions in class $\mathcal{F}$. We then have the following generalization guarantee:

**Theorem 2** (Excess Risk). *Suppose data $D$ contains of $N$ iid samples from distribution $\mathcal{D}$ and that the function $g$ has $\epsilon$ margin on data $D$ with probability $\mu$, i.e., $\text{Pr}_{x \sim D}[|g(x)| < \epsilon] < \mu$. Further, suppose that the class $\mathcal{C}_T$ has VC dimension $d$. Then, for $T \geq 4G_\infty^2 \ln 2N/\epsilon^2$, with probability $1 - \delta$ over the samples, the output $F_T$ of algorithm 1 satisfies:*

$$\text{err}(F_T) \leq \widehat{\text{err}}(g) + O\left(\sqrt{\frac{d\ln(N/d) + \ln(1/\delta)}{N}}\right) + \mu\,.$$

Note that the above theorem can easily be adapted to the case of margins and VC dimension of the class $\mathcal{C}_T$ being replaced with the corresponding fat-shattering dimensions. Furthermore, in the setting of stochastic minimax optimization, one can get population bounds directly by thinking of sample losses and gradients as stochastic gradients to the population objective. This is for example the view taken by [40]. In our work, we separate the population and sample bounds to simplify the presentation and the proofs.

# 5 Empirical Evaluation

We now evaluate B-DISTIL on both real-world and simulated datasets and over a variety of architecture types. We consider six real world datasets across three domains—vision, speech and sensor-data—as well as two simulated datasets. This allows us to evaluate our method on five architecture types: fully connected, convolutional, residual, densely connected networks and recurrent networks. Our code can be found at: `github.com/metastableB/bdistil`.

## 5.1 Dataset Information

For experiments with simulated data, we construct two datasets. The first dataset, referred to as *ellipsoid* is a binary classification dataset. Here the classification labels for each data point $x \in \mathbb{R}^{32}$ are determined by the value of $x^T A x$ for a random positive semidefinite matrix $A$. The second simulated dataset, *cube*, is for multiclass classification with 4 classes. Here labels are determined by distance to vertices from $\{-1, 1\}^{32}$ in $\mathbb{R}^{32}$, partitioned into 4 classes.

We also use six real world datasets for our experiments. Our image classification experiments use the *CIFAR-10*, *CIFAR-100*, *TinyImageNet* and *ImageNet* datasets. For time-series classification tasks we use the *Google-13* speech commands dataset. Here the task is keyword detection: given a one-second buffer of audio, we need to identify if any of 13 predefined keywords have been uttered in this. Finally, we use the daily sports activities (*DSA*) dataset for experiments with sensor data. Here the task is identifying the activity performed by an individual from a predefined set of sports activities, using sensor data. For detailed information of all datasets used see Appendix C.

## 5.2 Model Architecture Details

**Teacher models.** We use deep fully connected (FC) networks for classification on *Ellipsoid* and convolutional networks for *Cube*. For image classification on *CIFAR-10* and *ImageNet* dataset we use publicly available, pretrained ResNet models. We train reference DenseNet models for the

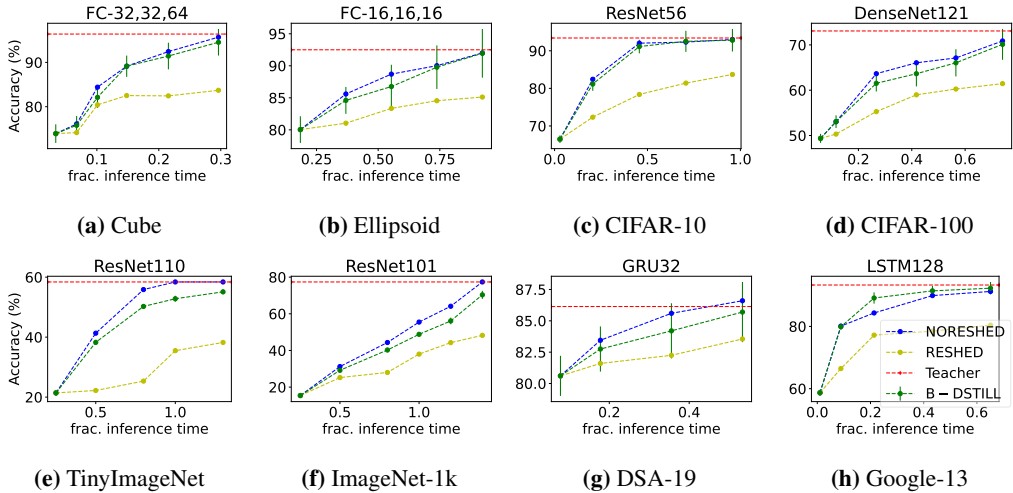

**Figure 2:** Accuracy vs. inference-time trade-offs. Inference time is reported as a fraction of teacher's inference time along with average ensemble accuracy and error bars. B-DISTIL performs this trade-off at runtime. The baseline NO-RESHED at inference time $\tau_w$ (x-axis) is the accuracy of a single model that is allowed $|\tau_w - 0|$ time for inference. Similarly the baseline RESHED at $\tau_w$ is the accuracy of an ensemble of models, where the model $w$ is allowed $|\tau_w - \tau_{w-1}|$ time to perform its inference. This is also the latency between the successive predictions from B-DISTIL. We can see that B-DISTIL (green) remains competitive to the oracle baseline (NO-RESCHED, blue) and outperforms weighted averaging (RESCHED, yellow).

*CIFAR-100* dataset based on publicly available training recipes (see Appendix C). As both spoken audio data (*Google-13*) and sensor-data (*DSA-19*) are time series classification problems, we use recurrent neural networks (RNNs). We train an LSTM-based architecture [24] on *Google-13* and a GRU-based architecture [10] on *DSA-19*. Except for the pretrained ResNet models, all other teacher models are selected based on performance on validation data.

**Student models.** For all distillation tasks, for simplicity we design the student base model class from the same architecture type as the teacher model, but start with significantly fewer parameters and resource requirements. We train for at most $T = 7$ rounds, keeping $\eta = 1$ in all our experiments. Whenever FIND-WL fails to find a weak learner, we expand the base class $\mathcal{F}$ using the connection specified as a hyperparameter. Since we need only at most $T = 7$ weak learners, we can pick small values of $R$ (say, 2). The details of the intermediate connections used for each dataset, hyperparameters such as the regularization parameter $\gamma$ in Equation 6 and hyperparameters for SGD can be found in Appendix C and D

### 5.3 Experimental Evaluation and Results

First, we present the trade-off between accuracy and inference time offered by B-DISTIL in the context of anytime inference and early prediction. We compare our models on top-1 classification accuracy and total floating point operations (FLOPs) required for inference. We use a publicly available profiler [35] to measure floating point operations. For simplicity of presentation, we convert these to the corresponding inference times ($\tau$) on a reference accelerator (NVIDIA 3090Ti).

**Anytime inference.** As discussed previously, in the anytime inference setting a model is required to produce a prediction even when its execution is interrupted. Standard model architectures can only output a prediction once the execution is complete and thus are unsuitable for this setting. We instead compare against the idealized baseline where we assume *oracle* access to the inference budget which is usually only available *after* the execution is finished or is interrupted. Under this assumption, we can train a set of models suitable various inference time constraints, e.g., by training models at various depths, and then pick the one that fits the current inference budget obtained by querying the oracle. We refer to this baseline as NO-RESHED and compare B-DISTIL to it on both synthetic and real world datasets in Figure 2. This idealized baseline can be considered an upper bound on the accuracy of B-DISTIL for a fixed inference budget.

| Dataset | Algorithm | Early-prediction Acc | | | | |
|---|---|---|---|---|---|---|
| | | $T = 50\%$ | | $T = 75\%$ | | $T = 100\%$ |
| | | Acc (%) | Frac | Acc (%) | Frac | Acc (%) |
| Google-13 | E-RNN | 88.31 | 0.48 | 88.42 | 0.65 | 92.43 |
| | B-DISTIL | 87.41 | 0.49 | 89.31 | 0.71 | 92.25 |
| DSA-19 | E-RNN | 83.5 | 0.55 | 83.6 | 0.56 | 86.8 |
| | B-DISTIL | 82.1 | 0.53 | 84.1 | 0.58 | 87.2 |

**Table 1:** Early prediction performance. Performance of the ensemble produced by B-DISTIL to the E-RNN algorithm [11]. The accuracy and the cumulative fraction of the data early predicted at $50\%, 75\%$ and $100\%$ time steps are shown. At $T = 100$, frac. evaluated is 1.0. The ensemble output by B-DISTIL with the early-prediction loss is competitive to the E-RNN algorithm. Unlike E-RNN, a method developed specifically for early prediction of RNNs, B-DISTILL is more generally applicable across model architecures and can also be used for offline.

B-DISTIL can improve on its initial prediction whenever inference jobs are allowed to be rescheduled. To contextualize this possible improvement, we consider the case where the execution is interrupted and rescheduled (with zero-latency, for simplicity) at times $\{\tau_1, \tau_2, \ldots, \tau_W\}$. We are required to output a prediction at each $\tau_w$. As an idealized baseline, assume we know these interrupt points in advance. One possible solution then is as follows: select models with inference budgets $|\tau_1|, |\tau_2 - \tau_1|, \ldots, |\tau_w - \tau_{w-1}|$. Sequentially evaluate them and at at each interrupt $\tau_w$, and output the (possibly weighted) average prediction of the $w$ models. We call this baseline RESCHED. Since the prediction at $\tau_w$ is a weighted average of models, we expect its performance to serve as a lower-bound for the performance of B-DISTIL. In the same figure (Figure 2) we compare B-DISTIL to RESHED.

We see that at all interrupts points in Figure 2, the predictions provided by B-DISTIL are competitive to that of the idealized baseline RESHED which requires the inference budget ahead of time for model selection, while being able to improve on its initial predictions if rescheduled. For instance, for the *CIFAR-100* dataset and at the interrupt point at $0.5$ on the $x$-axis, the predictions produced by B-DISTIL are comparable to a single model of the same inference duration, while being able to allow interrupts at all the previous points.

**Early prediction.**   To evaluate the applicability of our method for early prediction in online time-series inference, we compare the performance of B-DISTIL to that of E-RNN from [11]. Unlike B-DISTIL, which can be applied to generic architectures, E-RNN is a state-of-the-art method for early prediction that was developed specifically for RNNs. When training, we set the classification loss to the early-classification loss used in E-RNN training. We evaluate our method on the time-series datasets *GoogleSpeech* and *DSA*. The performance in terms of time-steps evaluated is compared in Table 1. Here, we see that B-DISTIL remains competitive to E-RNN for early prediction. Unlike E-RNN, B-DISTIL also offers early prediction for offline/batch evaluation time-series data. For such cases, a threshold can be tuned similar to E-RNN and B-DISTIL can evaluate the models in its ensemble in order of increasing cost, exiting when the prediction score crosses this threshold.

### 5.4   Training Considerations and Scalablility

**Connections.**   Our method uses intermediate connections to improve its performance. Although these connections are designed to be efficient, they still have an overhead cost over an averaging based ensemble. The FLOPs required to evaluate intermediate connections corresponding to the distillation tasks in Figure 2 is shown in Figure 3. Here, we compare the FLOPs required to evaluate the model from round $T$ to the FLOPs required evaluate the intermediate connections used by this model. Note that summing up all the FLOPs up to a round $T$, in Figure 3 gives the total FLOPs required to for the ensemble with the first $T$ models. For all our models, the overhead of connections is negligible when compared to the inference cost of the corresponding model.To evaluate the benefits offered by the intermediate connections, we can compare the results of B-DISTIL run with connections and B-DISTIL without connections. The latter case can be thought of as running the AdaBoost algorithm for distillation. Note that this is the same as the RESCHED baseline (weighted averaging).

On comparing the B-DISTIL plot in Figure 2 to the plot of RESCHED highlights the benefit of using intermediate connections. As in this work our focus is on finding weak learners in the presence of capacity gap, and we do not explore additional compression strategies like quantization, hard thresholding, low-rank projection that can further reduce inference cost.

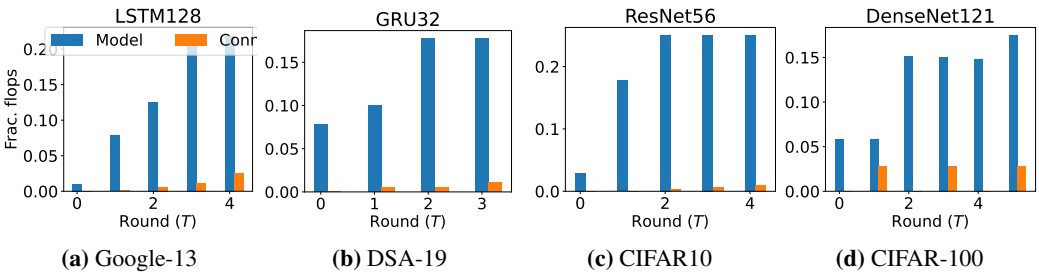

**(a)** Google-13      **(b)** DSA-19      **(c)** CIFAR10      **(d)** CIFAR-100

**Figure 3:** Overhead of connections. The floating point operations required to evaluate the model added in round $T$, compared to that required to evaluate just the connections used by this model. We present the results corresponding to datasets that have models with smaller required FLOPs overall. We see that even for these models the connections add relatively little overhead.

**Overheads of training/distillation.**   B-DISTIL requires additional compute and memory compared to a single model's training. Maintaining the two matrices $K_t^+$ and $K_t^-$ requires an additional $\mathcal{O}(NL)$ memory. Even for relatively large datasets with, say, $N = 10^6$ samples and $L = 1000$ classes, this comes out to a few gigabytes. Note that the two matrices can be stored in on disk and a batch can be loaded into memory for loss computation asynchronously using data loaders provided by standard ML frameworks. Thus the additional GPU memory requirement is quite small, $\mathcal{O}(bL)$ for a mini-batch size $b$, same as the memory required to maintain one-hot classification labels for a mini-batch. Since gradient computation is required only for the model being trained in each round, which typically is smaller than the teacher model, the backward pass is relatively cheap. For large datasets, loading the data for the teacher model forward pass becomes the bottleneck. See Appendix D for discussion on data loading, training time, and resource requirements for a ImageNet distillation.

**Sorting $\{\mathcal{F}\}_t$ using profiling.**   As mentioned in Section 3 B-DISTIL assumes the hypothesis classes in $\{\mathcal{F}\}_t$ are ordered on some metric, say, inference time. In practice, we achieve this at run-time by starting with a small base model, and profiling subsequent models considered in later rounds at run-time. For example, in PyTorch, we can use the `torch.autograd.profiler` module to profile the forward pass of a model. As a heuristic, we then sort the models in $\{\mathcal{F}\}_t$ based on the average inference time of the models in the ensemble.

## 6   Limitations, Broader Impact, and Future Work

In this paper we explore the problem of *progressive ensemble distillation*, in which the goal is to produce an ensemble of small weak learners from a large model, where components of the ensemble enable progressively better results as the ensemble size increases. To address this problem we propose B-DISTIL, an algorithm for progressive ensemble distillation, and demonstrate that it can be useful for a number of applications in efficient inference. In particular, our approach allows for a straightforward way to trade off accuracy and compute at inference time, and is critical for scenarios where inference may be interrupted abruptly or where variable levels of accuracy can be tolerated. We experimentally demonstrate the effectiveness of B-DISTIL by decomposing well-established deep models onto ensembles for data from vision, speech, and sensor domains. Our procedure leverages a stochastic solver combined with log barrier regularization for finding weak learners, use profiling for model selection and use intermediary connections to circumvent the issue of model capacity gap.

A key insight in this work is that posing distillation as a two player zero-sum game allows us to abstract away model architecture details into base class construction $\mathcal{F}$. This means that, conditioned on us finding a 'weak learner' from the base class, we retain the guarantees of the traditional boosting setup. Since these weak learners are only required to produce small improvements between rounds, we are able to reliably find such models. A caveat and potential limitation of this abstraction is that the user must design $\mathcal{F}$. Our research primarily focuses on on-device continuous inference, but our distillation procedure also holds benefits for cloud/data-center inference settings. This includes tasks like layer-fusion, load-balancing, and improved resource utilization, which merit further investigation in future studies. Finally, it is important to mention that our work prioritizes optimizing model accuracy while enabling compressed forms. However, another area for future research is exploring additional impacts of our approach on metrics such as fairness or robustness [25, 26, 32].

# 7 Acknowledgements

This work was supported in part by National Science Foundation Grants IIS2145670 and CCF2107024, a Meta Faculty Award, and the Private AI Collaborative Research Institute. Any opinions, findings, and conclusions or recommendations expressed in this material are those of the author(s) and do not necessarily reflect the National Science Foundation or any other funding agency.

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

# A Two-Player Minimax Games

In this section, we will look at the setting of two-player minimax games more closely.

Consider a class of hypotheses $\mathcal{F}$ and class of probability distributions $\mathcal{P}$. In addition, consider a loss function $\mathcal{L} : \mathcal{F} \times \mathcal{X} \to \mathbb{R}$ that is convex in its first argument. We consider two players whose pure strategy sets are $\mathcal{F}$ and $\mathcal{P}$ respectively. The loss and reward of the players is given by $F(f, \mu) = \mathbb{E}_{x \sim p}[L(f, x)]$ and the minmax value of the game is

$$\max_{p \in \mathcal{P}} \min_{f \in \mathcal{F}} F(f, \mu). \tag{9}$$

Note that this game is convex in the hypothesis player and concave in the distribution player. The objective of the game for the hypothesis player is trying to find a hypothesis that has low loss on the worst case distribution from class $\mathcal{P}$. Conversely, the distribution player is trying to construct a distribution that is as hard as possible for the hypothesis player to learn.

Also, note that under reasonable conditions on $\mathcal{F}$ and $\mathcal{P}$, we have the minimax theorem holds (see Chapter 6, Schapire and Freund [39]),

$$\max_{p \in \mathcal{P}} \min_{f \in \mathcal{F}} \mathbb{E}_{x \sim p}[L(f, x)] = \min_{f \in \Delta(\mathcal{F})} \max_{p \in \mathcal{P}} \mathbb{E}_{x \sim p}[L(f, x)]. \tag{10}$$

Here, $\Delta(\mathcal{F})$ is the set distributions over functions $\mathcal{F}$. From this, we can see that as long as we are allowed to aggregate functions from the base class, we have can do as well as we could if we had access to the distribution

The interesting algorithmic question would be to find a distribution over hypotheses that achieves the minimum above. Note that the loss function is stochastic and thus, we need to formalize the access the player has to the loss function. We will formulate this in the following stochastic way.

**Definition 2.** *An algorithm* ORACLE *is said to be a* $(\beta, \delta)$ *weak-gradient if*

$$\langle \text{ORACLE}(f, p), \nabla_f F(f, p) \rangle \geq \beta \tag{11}$$

*with probability* $1 - \delta$.

Here $\nabla_f$ denotes the functional gradient of $F$. This notion is similar to the weak learning assumptions usually used in the boosting literature. Given such an oracle one can ask for methods similar to first order methods for convex optimization, such as gradient descent, to solve the minimax problem. These algorithms iteratively maintain candidate solutions for the both the players and update each of these using feedback from the state of the other player. In our particular setting, the hypothesis player updates using the vector $h$ in eq. (3).

**Motivating Example.** Let $\mathcal{F}$ be a class of hypotheses, let $\mathcal{P}$ is the set of all distributions over a finite sample set $\{x_1, \ldots, x_n\}$ and let $L$ be the 0-1 loss. Note that in this setting, the oracle from Definition 2 is analogous to weak learning. In this setting, from the minimax theorem and the existence of a weak learner, we get that there is a single mixture of hypothesis of $\sum_i \alpha_i f_i$ such that loss under every distribution in $\mathcal{P}$ which corresponds to zero training error. Thus we can think of boosting algorithms as approximating this minmax equilibrium algorithmically. Similarly, the weak learning condition in [40] is similar in spirit to the condition above.

With the condition from Definition 2, one can consider many frameworks for solving minimax games. One general technique is to consider two no-regret algorithms for online convex optimization to play against each other. Let us briefly look at the definition of regret in this setting.

**Definition 3** (No-Regret)**.** *Let* $K, A$ *be convex sets. At each time, a player observes a point* $x_t \in K$ *and chooses an action* $a_t \in A$. *The regret of the algorithm is defined as*

$$R_T = \max_{a \in A} \sum_{t=1}^{T} \langle a, x_t \rangle - \sum_{t=1}^{T} \langle a_t, x_t \rangle. \tag{12}$$

Online learning is a well-studied area of machine learning with a rich set of connections to various areas in mathematics and computer science. In particular, there are frameworks in order to construct algorithms such as follow-the-perturbed leader, follow-the-regularized leader and mirror descent.

Our algorithm can be seen as a version of mirror descent with the entropy regularizer and Theorem 1 as a version of the regret guarantee for the algorithm. In addition to those mentioned above, there are several other frameworks considered to solve minimax games such as variational inequalities, extragradient methods, optimistic methods, etc. We believe this general framework is a useful one to consider for many learning tasks, especially in settings where we have function approximation.

# B  Proofs of Main Theorems

Here, we provide a proof of Theorem 1, which is restated below:

**Theorem.** *Suppose the class $\mathcal{F}$ satisfies that for all $f \in \mathcal{F}$, $\|f - g\|_\infty \leq G_\infty$. Let $F = \{f_t\}$ be the ensemble after $T$ rounds of Algorithm 1, with the final output $F_t = \frac{1}{T} \sum_{t=1}^{T} f_t$. Then for $T \geq \ln 2N$ and*

$$\eta = \frac{1}{G_\infty} \sqrt{\frac{\ln 2N}{T}}$$

*we have for all $j$*

$$\|F_{t,j} - g_j\|_\infty \leq G_\infty \sqrt{\frac{\ln 2N}{T}} - \frac{1}{T} \sum_{t=1}^{T} \gamma_t(j)$$

*where $F_{t,j}$ and $g_j$ are the $j^{th}$ coordinates of the functions $F_t$ and $g$ respectively.*

*Proof.* For simplicity, we assume that $f_t$ and $g$ are scalar valued functions, since the proof goes through coordinate-wise. At each time, define the edge of the weak learning algorithm to be

$$\gamma_t = \sum_i K_t^+(i)(f_t(x_i) - g(x_i)) + \sum_i K_t^-(i)(g(x_i) - f_t(x_i))$$

Let $Z_t$ denote the normalizing constant at time $t$, that is,

$$Z_t = \sum_i K_t^+(i) \exp\left(-\eta\left(f_t(x_i) - g(x_i)\right)\right) + K_t^-(i) \exp\left(\eta\left(f_t(x_i) - g(x_i)\right)\right)$$

From the update rule, we have

$$
\begin{aligned}
K_{T+1}^+(i) &= \frac{K_T^+(i) e^{\eta(f_T(x_i) - g(x_i))}}{Z_T} \\
&= \frac{K_1^+(i) \exp\left(-\eta \sum_{t=1}^{T} \left(f_t(x_i) - g(x_i)\right)_j\right)}{\prod_{t=1}^{T} Z_t} \\
&= \frac{K_1^+(i) \exp\left(-\eta T (F_T(x_i) - g(x_i))\right)}{\prod_{t=1}^{T} Z_t}
\end{aligned}
$$

and similarly

$$K_{T+1}^-(i) = \frac{K_1^-(i) \exp\left(\eta T (F_T(x_i) - g(x_i))\right)}{\prod_{t=1}^{T} Z_t}$$

First, we bound $\ln(Z_t)$:

$$
\begin{aligned}
\ln(Z_t) &= \ln\left(\sum_i K_t^+(i) \exp(-\eta(f_t(x_i) - g(x_i))) + \sum_i K_t^-(i) \exp(\eta(f_t(x_i) - g(x_i)))\right) \\
&\leq \ln\left(\sum_i K_t^+(i)\left(1 - \eta(f_t(x_i) - g(x_i)) + \eta^2(f_t(x_i) - g(x_i))^2\right)\right. \\
&\qquad\qquad \left. + \sum_i K_t^-(i)\left(1 + \eta(f_t(x_i) - g(x_i)) + \eta^2(f_t(x_i) - g(x_i))^2\right)\right) \\
&\leq \ln\left(1 - \eta \sum_i K_t^+(i)(f_t(x_i) - g(x_i)) + \eta \sum_i K_t^-(i)(f_t(x_i) - g(x_i)) + \eta^2 G_\infty^2\right) \\
&\leq -\eta\gamma_t + \eta^2 G_\infty^2
\end{aligned}
$$

where the second step follows from the identity $\exp(x) \leq 1 + x + x^2$ for $x \leq 1$, provided that $\eta \leq \frac{1}{G_\infty}$. This gives us a bound on regression error after $T$ rounds:

$$-\eta T \left(F_T(x_i) - g(x_i)\right) = \ln(K_{T+1}^+(i)) - \ln(K_1^+(i)) + \sum_{t=1}^{T} \ln(Z_t)$$

$$\leq \ln\left(\frac{K_{T+1}^+(i)}{K_1^+(i)}\right) + \sum_{t=1}^{T} -\eta\gamma_t + \eta^2 G_\infty^2$$

$$= \ln\left(\frac{K_{T+1}^+(i)}{K_1^+(i)}\right) + \eta^2 T G_\infty^2 - \eta \sum_{t=1}^{T} \gamma_t$$

$$\leq \ln 2N + \eta^2 T G_\infty^2 - \eta \sum_{t=1}^{T} \gamma_t ,$$

where the last bound follows since $K_1^+ = \frac{1}{2N}$ and $K_{T+1}^+ \leq 1$. Similarly, we have the bound

$$\eta T \left(F_T(x_i) - g(x_i)\right) \leq \ln 2N + \eta^2 T G_\infty^2 - \eta \sum_{t=1}^{T} \gamma_t$$

Combining the two equations we get that

$$\sup_i |F_T(x_i) - g(x_i)| = \|F_T - g\|_\infty \leq \frac{\ln 2N}{\eta T} + \eta G_\infty^2 - \frac{1}{T} \sum_{t=1}^{T} \gamma_t .$$

If we choose $\eta = \frac{1}{G_\infty}\sqrt{\frac{\ln 2N}{T}}$ to minimize this expression, then we get the following bound on regression error:

$$\|F_t - g\|_\infty \leq -\frac{1}{T} \sum_{t=1}^{T} \gamma_t + G_\infty \sqrt{\frac{\ln 2N}{T}} .$$

which is exactly Equation (8). Note that the value of $\eta$ only satisfies the condition $\eta \leq \frac{1}{G_\infty}$ when $T \geq \ln 2N$, which is the time horizon after which the bound holds. This finishes the proof of Theorem 1. □

Now, we provide a proof of Theorem 2 which follows from the VC dimension bound and Theorem 1. Before we begin, we setup some notation. Given a function $f$, distribution $\mathcal{D}$ over space $\mathcal{X} \times \mathcal{Y}$ where $\mathcal{X}$ is the input space and $\mathcal{Y}$ is the label space, and data $D$ consisting of $N$ iid samples $(x, y) \sim \mathcal{D}$, we define

$$\widehat{\mathrm{err}}(f) = \Pr_{(x,y)\sim D}[\mathrm{sign}(F_T(x) \neq y)] \qquad \mathrm{err}(f) = \Pr_{(x,y)\sim \mathcal{D}}[\mathrm{sign}(F_T(x) \neq y)]$$

**Theorem** (Excess Risk). *Suppose data $D$ contains of $N$ iid samples from distribution $\mathcal{D}$. Suppose that the function $g$ has large margin on data $D$, that is*

$$\Pr_{x\sim D}[|g(x)| < \epsilon] < \mu$$

*Further, suppose that the class $\mathcal{C}_T$ has VC dimension $d$, then for*

$$T \geq \frac{4G_\infty^2 \ln 2N}{\epsilon^2},$$

*with probability $1 - \delta$ over the draws of data $D$, the generalization error of the ensemble $F_T$ obtained after $T$ round of Algorithm 1 is bounded by*

$$\mathrm{err}(F_T) \leq \widehat{\mathrm{err}}(g) + O\left(\sqrt{\frac{d\ln(N/d) + \ln(1/\delta)}{N}}\right) + \mu$$

*Proof.* Recall the following probability bound Schapire and Freund [39, theorem 2.5] which follows Sauer's Lemma:

$$\Pr\left[\exists f \in \mathcal{C}_T : \text{err}(f) \geq \widehat{\text{err}}(f) + \epsilon\right] \leq 8 \left(\frac{me}{d}\right)^d e^{-m\epsilon^2/32}$$

which holds whenever $|D| = N \geq d$. It follows that with probability $1 - \delta$ over the samples, we have for all $f \in \mathcal{C}_{\mathcal{T}}$

$$\text{err}(f) \leq \widehat{\text{err}}(f) + O\left(\sqrt{\frac{d\ln(N/d) + \ln(1/\delta)}{N}}\right) \tag{13}$$

Since we choose $T = \frac{4G_\infty^2 \ln 2N}{\epsilon^2}$, by Theorem 1, we have

$$\forall x \in D : \|F_t - g\|_1 \leq G_\infty \sqrt{\frac{\ln 2N}{T}} \leq \frac{\epsilon}{2}$$

Since $g$ has $\epsilon$ margin on data with probability $1 - \mu$, we have

$$\widehat{\text{err}}(F_t) \leq \widehat{\text{err}}(g) + \mu \tag{14}$$

Combining eqs. (13) and (14), we get

$$\text{err}(F_T) \leq \widehat{\text{err}}(g) + O\left(\sqrt{\frac{d\ln(N/d) + \ln(1/\delta)}{N}}\right) + \mu$$

which completes the proof. $\qquad\square$

## C  Dataset Information and Training Recipe

We use six publicly available real world datasets in our experiments. The train-test splits for all the dataset as well as the sources are listed here:

### C.1  Dataset

| Dataset | Train-samples | Test/Val-samples | Num.-labels | Source |
|---|---|---|---|---|
| CIFAR-10 | 50000 | 10000 | 10 | [29] |
| CIFAR-100 | 50000 | 10000 | 100 | [29] |
| DSA-19 | 6800 | 2280 | 19 | [14] |
| Google-13 | 52886 | 6835 | 13 | [43] |
| ImageNet-1k | 1281167 | 50000 | 1000 | [37] |
| TinyImageNet-200 | 100000 | 10000 | 200 | [30] |

We use two synthetic datasets in our experiments, *ellipsoid* and *cube*. To construct the ellipsoid dataset, we first sample a $32 \times 32$ matrix $B$, each entry sampled *iid*. We define $A := B^T B$ as our positive semi-definite matrix, and $I[x^T A x \geq 0]$ determines the label of a data point $x$. We sample 10k points uniform randomly from $[-1, 1]^{32}$ and determine their labels to construct our data sample. We randomly construct a 80-20 train-test split for our experiments.

To construct *cube*, we first sample 16 vertices uniform randomly from $[-1, 1]^{32}$ and split them into 4 equal sets, say $\{S_1, \ldots, S_4\}$. As before, we sample 10k points uniformly from $[-1, 1]^{32}$ and determine the label $y(x)$ of each point $x$ based on the closest vertex in $\{S_1, \ldots, S_4\}$.

$$y(x) = \arg\min_i \min_{x' \in S_i} \|x - x'\|.$$

### C.2  Training Recepies

We use stochastic gradient descent (SGD) with momentum for all our experiements. For experiments on CIFAR100 and CIFAR10, we use a learning rate of $0.1$, a momentum paramter of $0.9$, and weight decay of $5 \times 10^{-4}$. We train for 200 epochs and reduce the learning rate by a factor of $0.2$ in after

| Teacher model | Residual Blocks | Embedding dims | Strides |
|---|---|---|---|
| ResNet56 | 1 | 8 | 1 |
| | 2,2 | 8,8 | 1,1 |
| | 2,2 | 16,16 | 1,2 |
| | 2,2,3 | 16,32,64 | 1,2,2 |

**Table 2:** Base model configuration used for ResNet56 distillation on CIFAR-10.

| Teacher model | Blocks | growth-rate |
|---|---|---|
| DenseNet121 | 4, 8 | 12 |
| | 4, 8, 8 | 6 |
| | 8, 16, 12 | 6 |

**Table 3:** Configuration used for DenseNet121 distillation on CIFAR-100.

30%, 60% and 90% of the epoch execution. We perform a 4-GPU data-parallel training for ImageNet with a per-gpu batch size of 256, learning rate 0.1, momentum 0.9, regularization $\gamma$ of 1.0, and a weight decaur of $1e - 4$. We train for 90 epochs with and discount the learning rate by a factor of 0.1 at 30% and 60% epochs. For experiments with time series data, Google-13 and DSA-19, we use a fixed learning rate of 0.05 and a momentum of 0.9. We do not use weight decay or learning rate scheduling for time-series data.

### C.3   Profiling Based Model Selection

To estimate execution latency, we leverage a third-party library, such as the *Deep Speed* framework [35], which enables us to measure the total floating-point operations required for inference. By randomly initializing a single sample with a batch size of 1, we obtain the FLOPs values for various real-world datasets. This profiling process serves as a reliable indicator of real-world performance, allowing us to rank the models based on their profiles. Notably, the profiling results only need to hold relative to other candidate models. Furthermore, we can utilize existing profiling models, such as ARM Fast Models for ARM-based mobile platforms, QEMU for x86 platforms, and NVIDIA NSight's GPU emulators for GPU platforms, to estimate the performance on different hardware architectures. However, it's important to note that these software-based solutions provide approximate performance estimates, and accurate evaluation of real performance requires access to the actual hardware. We are not restricted to latency based rankings. In fact, other useful metrics like throughput in sample processed per second, can also be used for candidate model selection.

## D   Base Model Configuration

The base class configurations used for all our experiments in Figure 2 is provided in Tables 2, 3, 6 and 7. Note that we use standard model architectures implemented in Pytorch and the parameters correspond to the corresponding function arguments in these implementation. We use pretrained models from the *torchvision* library.

| Teacher model | Residual Blocks | Embedding dims | Strides |
|---|---|---|---|
| ResNet110 | 2, 2 | 16, 16 | 1,2 |
| | 2, 3 | 32, 64 | 1,2 |
| | 2, 2, 3 | 16, 32, 64 | 1,1,2 |
| | 2, 3, 3 | 16, 32, 64 | 1,2,2 |

**Table 4:** Base models for TinyImageNet.

| Teacher model | Residual Blocks | Embedding dims | Strides |
|---|---|---|---|
| | 2, 2 | 16, 16 | 1,2 |
| ResNet101 | 2, 3 | 32, 64 | 1,2 |
| | 2, 2, 3 | 16, 32, 64 | 1,1,2 |
| | 2, 3, 3 | 32, 32, 64 | 1,2,2 |

**Table 5:** Base models for ImageNet.

| Teacher model | hid. dims. |
|---|---|
| | 4,4 |
| LSTM128 | 16,8 |
| | 20,12 |
| | 20,32 |

**Table 6:** Configuration used for LSTM128 distillation on Google-13.

| Teacher model | hid. dims. |
|---|---|
| | 4,4 |
| GRU32 | 8,16 |
| | 16,16 |
| | 32,16 |

**Table 7:** Configuration used for GRU32 distillation on DSA-19.

# E   Additional Results

## E.1   Connections and parallel execution schedules.

Our focus in this work has been sequential execution of the models. While reusing previously computed features is clearly beneficial for finding weak learners in this setup, the presence of connections across models prevent them from being scheduled together for execution whenever otherwise possible. To manage this trade off between parallelization and expressivity, we try to restrict the number of connections to at most one between models, and further restrict the connection to later layers of the model. Connections in the later layers of networks impose fewer sequential blocks in inference and allows for better parallelization.

Let $\phi_{t,l}(x)$ denote the activation produced at layer $l$ by the weak learner at round $t$, on data-point $x$. Then some of the connections we consider are

- $\phi_{(t,l)}(x) - \phi_{(t+1,l)}(x)$, to learn the error in features at layer $l$.
- $\phi_{(t,l)}(x) + \text{id}(x)$, standard residual connection at layer $l$.
- $\phi_{(t+1,l)}[\phi_{(1,l)}(x), \ldots, \phi_{t,l}(x)]$, dense connections at layer $l$ across rounds.
- Simple accumulation operations.
- Recurrent networks: LSTM and GRU.

## E.2   Training ImageNet and Throughput Optimization

Large datasets like ImageNet require additional care to ensure good resource utilization. Since the backward pass of our training only involve a single student model, it can be computed quite efficiently even for large datasets. This is particularly true in earlier rounds. For such cases we use simple producer consumer architecture where a GPU is dedicated to producing and queuing data batches and target predictions produced by the teacher. Training routines running on separate GPUs consume these batches for their training.

As mentioned in Section 3, since B-DISTIL only cares that it is provided with an ordered list of candidates, we can modify the weak learning finding step to prefer weak learners that provide higher inference throughput, measured as samples processed per second. Throughput metric is more meaningful in the cloud inference deployment setting where batched inference is the norm. The ability to trade a small performance for the number of inference requests to process can be useful in such settings. Implementation details can be found in the included code base. We note that we do not optimize for throughput and inference latency simultaneously in this work, but is an interesting direction of future research.

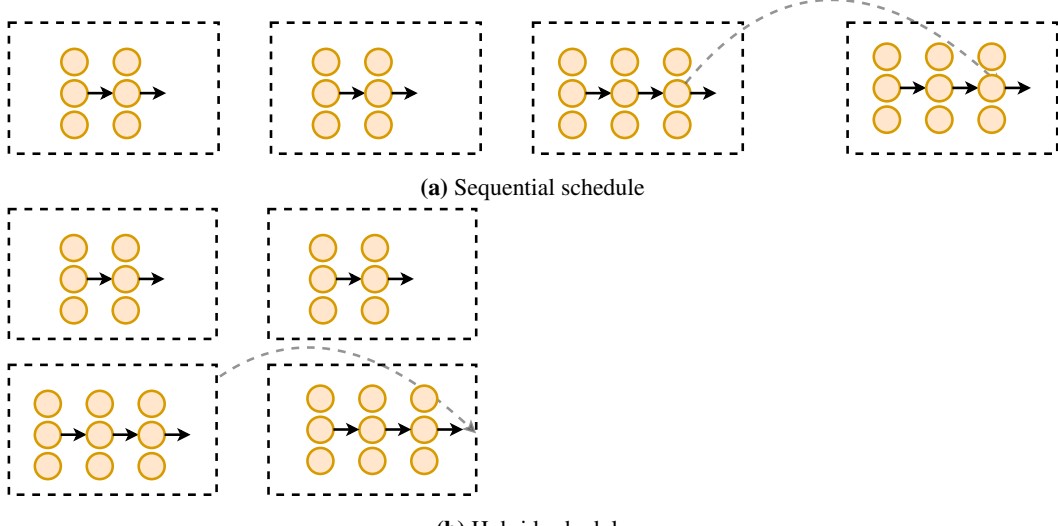

**(a)** Sequential schedule

**(b)** Hybrid schedule

**Figure 4:** A schematic description of a sequential execution scheme for an ensemble of four models, being evaluated one after the other from left to right. The last model in the ensemble reuses the actions of the previous one, causing a blocking dependency. Thus we cannot trivially execute all models in parallel. However, since the connection is between the last layers of the network, we can construct hybrid execution schemes as in (b). Here, pairs of models are executed together.

