# OpenReview forum: "Progressive Ensemble Distillation: Building Ensembles for Efficient Inference"
_NeurIPS.cc/2023/Conference — NeurIPS 2023 poster_

### Official Review · Reviewer_HW4G · 2023-06-20

**Soundness:** 3 good
**Presentation:** 3 good
**Contribution:** 3 good
**Rating:** 6
**Confidence:** 3

**Summary:**

The paper addresses the problem of obtaining an ensemble of small models suitable for flexible inference requirements and anytime inference, somewhat similar to cascading classifiers. A key contribution is the derivation of a weak learning condition for the distillation of a pre-trained to an ensemble of smaller students as well as an algorithm to obtain such an ensemble. The students are allowed to reuse intermediate activations of other students to efficiently expand the student model hypothesis set. The method is supported by theoretical results on the generalization error, and empirical results on classification tasks for both synthetic, vision, and sensor data.

**Strengths:**

- The paper is well-written, clear, and well-organized. It is mostly easy to follow and understand the argumentation.
- The proposed technique is supported both theoretically and empirically, which is a strong feature.

**Weaknesses:**

- The computational requirements during training are unclear. Since Algorithm 2 requires the fitting of potentially multiple models to obtain each student it appears to potentially be quite expensive (especially with large $R$ and/or large `max-search`), but this is not addressed explicitly in the paper
- The dimensions of $K_t^-$ and $K_t^+$ are $N \times L$ and especially for large $N$, this could be a bottleneck during training (granted, for a mini-batch this is less severe). Furthermore, while reusing stored activations of previous students in subsequent students might keep the parameter count stable, it also requires additional memory and carefulness in which activations to store. Thus there is some overhead on storing and loading activations and $K$-matrices throughout training.
- The empirical results are weak at comparing to other baseline methods, and the method is struggling at the TinyImageNet and ImageNet-1k tasks, where additional inference time is required compared to the teacher.

Minor:
- L14/15: Claiming distillation is a rigid procedure seems too bold, as a multitude of distillation techniques exists providing options to obtain lots of different students. Granted, most are aimed at obtaining a single student, but distillation in general is very flexible.
- L121: "proabability" -> "probability"
- Algorithm 1: $R$ is not specified in the algorithm, but needs deduction from Section 3.3
- Inconsistent use of RESCHED and RESHED
- Formatting of B-Distill and E-RNN are inconsistent in different places in the paper (e.g. L300-309).



**Questions:**

Generally, my largest concern is the lack of other baseline methods for the empirical results and the unclear computational requirements during training.

Additionally:
- In Section 5.3 (and Figure 2) it appears that FLOPs are converted to inference time, but it is unclear how, and if this conversion actually holds for an actual implementation. Consider measuring the actual inference time instead.
- Clarify the computational requirements during training. What is the overhead on memory and training speed when storing and loading activations and $K$-matrices?
- Include common distillation techniques and other methods for early-exit or anytime inference as baselines. E.g. it is unclear how NO-RESHED compares to other distillation schemes with appropriately sized students, or if existing anytime inference techniques surpass the proposed.

Minor:
- Some dimensions of e.g. $x_i$ and $f(x_i)$ are not clear from the paper (but can be deduced), and e.g. in L119, what indices are $j$ summing over? Consider introducing dimensions more clearly early on.
- Figure 2: Since the $x$-axis is the fraction of teacher inference time, the teacher should be marked by a dot and not a line, since the teacher is not able to perform inference at every possible inference time. It should be clear that the teacher is not flexible in inference time.

**Limitations:**

Sufficiently addressed.

---

> ### Author Rebuttal · Authors · 2023-08-09
>
> Thank you for your constructive feedback and questions.
>
> **Inference time calculations.** The inference time numbers are end-to-end numbers reported by the DeepSpeed profiler on NVIDIA 3080Tis. The No-RESCHED baseline is used as an idealized baseline, and a modified version trained using distillation will improve on the numbers reported now. In both cases, we expect this baseline to be better than B-DISTIL. We use the training based baseline instead of the distillation as the teacher model is a natural case of baseline at `frac_inf = 1.0`.
> Large values of `max-search` could indeed lead to an increase in training times if FIND-WL fails often, which is not typically a concern for larger models (later rounds). Models in earlier rounds are often relatively small, and we do see additional time required for training as for these models the data movement often becomes the training bottleneck.
>
> **Scalability and compute requirements.** Although B-DISTIL takes additional inference time to make accurate predictions for ImageNet, the teacher models in this case have 100+ layers. Tasks in efficient inference that rely on smaller models for image datasets, or scenarios (e.g., edge inference) involving sensor/audio data streams, are key application areas that stand to benefit from B-DISTIL. However, even at larger scales, where the data distribution during inference is skewed towards 'easier' samples, B-DISTIL can be advantageous in the average case, as the majority of the predictions can be completed quickly (Google-13, Table-1, T=50%). In fact our goal in providing the ImageNet scale experiments was to demonstrate scalability of our implementation, especially since maintaining and updating the weight-matrix of a 1000 class, 1M training image dataset can be non-trivial. We will clarify this in the main text and add a discussion about the techniques we use to manage compute requirements (example, streaming weights off-disk asynchronously [code: `ddist.data:DataFlowControl`], performing weight updates in log-space [code: `ddist:ClfPlayer.log_space_update()`], using a shared-memory object store [code: `ddist.dispatch` utilizing `ray` backend]) in the appendix.
>
> **Baseline methods.** The NO-RESCHED baseline is trained using standard distillation on the same model structure used by B-DISTIL. If we were to compare the models at fixed $T$, a single model trained using standard distillation techniques does outperform B-DISTIL (Figure 2) especially for later rounds. However, this baseline cannot be deployed in practice for anytime inference as it requires knowledge of available inference time upfront to pick the ‘right model’. We demonstrate that B-DISTIL remains competitive to this baseline while being realizable in practice. As for early prediction, the E-RNN method we compare against (Table 1) is a standard method used for early prediction in sequential inference. We will clarify these points in the main draft.
>
>
> **Minor.** Thank you for the suggestion on the horizontal line for the teacher model and other corrections/typos, which we will carefully address in our revision. We will also revise the draft to use only one unit of measurement, and improve and move details about the profiling step to the main draft. Thanks very much for these suggestions to improve presentation.

---

> > ### Comment · Reviewer_HW4G · 2023-08-11
> >
> > You write *"The No-RESCHED baseline is used as an idealized baseline, and a modified version trained using distillation will improve on the numbers reported now."*, but later also write *"The NO-RESCHED baseline is trained using standard distillation on the same model structure used by B-DISTIL"*. I get the intuition behind the NO-RESHED baseline (and why it is a difficult baseline), but it should be clearer what the actual training procedure is and/or what architectures are used for each time.

---

> > > ### Author Response · Authors · 2023-08-13
> > > **Response to comment by HW4G**
> > >
> > >
> > >  We apologize for the confusion. When we say the NO-RESCHED baseline is trained using distillation on “the same model structure as the teacher model”, what we mean is that if the teacher model is based on ResNet architecture then the student model in this baseline will also be from the same family. The model configuration  for the student model at a specific round $T$, for instance the number of layers, number of blocks, etc are chosen so that the inference time is comparable to the ensemble of models produced by B-DISTIL at the end of the same round. Standard distillation is used to train this student model (i.e., training against soft logits). However, instead of considering a single deep ResNet model of appropriate size, we could also consider randomly re-initializing and re-training all the parameters in the ensemble produced by B-DISTIL. As we mention, we include the former baseline as the teacher model is a data point on this plot at `frac_infr=1.0`. Moreover, for a specific $T$, a single ResNet model is denser and deeper (less capacity gap) when compared to the ensemble structure; while the ensemble structure at $T$ has a similar compute requirement, it typically contains relatively shallower models which _could_ cause a drop in performance. We will add a clarification of this point in the main text and specify the exact model configuration used for this baseline in the appendix.

---

> > > > ### Comment · Reviewer_HW4G · 2023-08-14
> > > >
> > > > I appreciate the clarification from the authors and suggest including the clarification as mentioned. However, I stand by my current score of accepting the paper.

---

### Official Review · Reviewer_nVQk · 2023-07-05

**Soundness:** 3 good
**Presentation:** 3 good
**Contribution:** 3 good
**Rating:** 6
**Confidence:** 2

**Summary:**

This paper studies the problem of "progressive distillation": Given a large teacher model, the task is to decompose into smaller student model so that progressively evaluating additional models in this ensemble results into more accurate predictions.

Τhe main contributions of this paper are:

(i) A principled approach called B-DISTIL for approaching the progressive distillation problem: The authors formulate a two player zero-sum game, from which they derive a weak learning condition. B-DISTIL approximately solves this game.
(ii) Theoretical guarantees for the proposed approach under certain assumptions.

**Strengths:**

Principled approach with theoretical guarantees that seems to perform well in real-world settings.

**Weaknesses:**

The proposed approach seems somewhat sophisticated — perhaps not very easy to implement even. By reading the paper, it was not clear to me whether there exists a simpler (but non-idealized) baseline that could be used for comparison — mostly to reassure the reader that the introduced sophistication is actually necessary.

**Questions:**

Is there a simple (maybe standard Knoweledge-Distillation-based) baseline that is meaningful to compare against?

For example, in Lines 70-72 the authors mention that "when performing distillation onto a weighted combination of ensembles, it has been observed that adding additional models into the ensemble does not dramatically improve performance over that of a single distilled model".

While this could be the case, could such an approach be used as a simple baseline for this setting, so that one could see what are the trade-offs between implementing a simple approach and potentially improving performance by implementing a more sophisticated one like the one proposed by the authors?  (I understand if the answer is "there's no simple way to approach this problem", but maybe then perhaps this should be mentioned more explicitly.)


**Limitations:**

The authors have explained the limitations of their work.

---

> ### Author Rebuttal · Authors · 2023-08-09
>
> Thank you for your helpful comments and feedback.
>
> Although our algorithm seems sophisticated, we note that most of the additional sophistication (on top of standard boosting) is restricted to FIND-WL sub-routine. Conceptually, the high-level algorithm has the same flavor as many boosting methods, where the existence of a subroutine equivalent to `FIND-WL` is assumed without specifying details.  However, scaling to large training datasets is non-trivial even for standard boosting due to the weight-matrices. The ImageNet scale experiments demonstrate the scalability of our implementation, where we use various techniques (example, streaming weights off-disk asynchronously [code: `ddist.data:DataFlowControl`], performing weight updates in log-space [code: `ddist:ClfPlayer.log_space_update()`], using a shared-memory object store [code: `ddist.dispatch` utilizing the `ray` backend]) to manage compute requirements.
>
> Regarding baselines, a simple baseline that only uses traditional distillation is the baseline consisting of many small distilled models, sequentially evaluated till interrupted. We have included this baseline, which we call "RESCHED", in our experiments (Figure 2). Generally, such an approach, which is basically a weighted combination of distilled models, is known to be not very effective. We will make this more explicit in the main draft.

---

> > ### Comment · Reviewer_nVQk · 2023-08-13
> >
> > Thank you for your response. I stand by my original (positive) assessment.

---

### Official Review · Reviewer_85M5 · 2023-07-06

**Soundness:** 3 good
**Presentation:** 3 good
**Contribution:** 2 fair
**Rating:** 5
**Confidence:** 3

**Summary:**

This paper proposes B-DISTILL, a progressive distillation algorithm that allows for easy trade-off between accuracy and inference-time/latency at runtime. By modeling knowledge distillation as a zero-sum game problem, B-DISTILL utilizes the intermediary connection modules to train and aggregate the sub-student models progressively, resembling the traditional boosting methods. The paper provides mathematical proofs that guarantee the convergence and generalization of B-DISTILL. The experimental results demonstrate the efficiency of B-DISTILL in both anytime inference and early prediction tasks.

**Strengths:**

1. The paper presents a novel perspective by redefining the knowledge distillation problem and effectively applying it to the tasks of anytime inference and early prediction.
2. The paper provides complete mathematical proof and experimental validation to support its claims.

**Weaknesses:**

1. While the method proposed in this paper introduces a novel perspective, its application scope and advantages appear to be quite limited.
2. It seems that some dynamic network structures could potentially be used to address the anytime reference problem. However, it appears that the paper lacks a comparative analysis with relevant methods in terms of results.
3. It might be worth considering modifying the title. B-DISTILL is more like a training method specifically designed for efficient inference rather than a knowledge distillation-related approach.

**Questions:**

1. I would like to know the role of knowledge distillation in this context. Why not directly use ground truth (gt) as the fitting target?
2. What results would be obtained if the teacher model is directly replaced with softened labels?

**Limitations:**

Yes.

---

> ### Author Rebuttal · Authors · 2023-08-09
>
> Thank you for constructive feedback and questions, which we hope we have addressed in our response below.
>
> **Application scope.** We kindly disagree with the reviewer that the scope and advantages of our work are quite limited. Due to resource constraints in efficient inference applications it is common to leverage small to medium scale models as a starting point, e.g., models with fewer than hundred layers for image tasks, or small models for sensior/audio data streams (see, e.g., [1, 2]). These are applications/scenarios that stand to benefit significantly from the flexibility and efficiency enabled by our approach, B-DISTIL. However, even at larger scales, where the inference time data distribution is skewed towards 'easier' samples, our results show that B-DISTIL can be advantageous in the average case, as the majority of the predictions can be completed quickly (see for example Google-13, Table-1, T=50%).
>
> **Dynamic Networks.** We thank the reviewer for pointing out dynamic networks, which is a broad term that can refer to many potential approaches for dynamically adjusting model structure or parameters (see e.g., the survey [3]). At a high level these are categorized into methods that depend on input samples, training procedure, or inference procedure. The most related works to ours are those that depend on samples and dynamically adjust the model architecture (e.g., early exit, layer skipping schemes). Relative to existing methods from this category, our work provides a principled way of performing decomposition in the presence of large capacity gaps and early exit requirements, which can apply to both anytime inference and early exit problems, and is not tied to a specific modality (e.g., image, language). We have already discussed some of the most related approaches from the broad class of dynamic networks in our related work section (e.g., Huang et al. 2018, Ruiz and Verbeek, 2020 (HNE)) and have compared empirically to E-RNN, a representative approach, but will include a broader discussion that better positions these methods in the context of dynamic networks more generally.
>
> **Role of distillation.** We employ distillation instead of just the ground-truth labels to make learning with smaller capacity models possible. The temperature smoothening step of distillation combined with directly optimizing on the results aids our training procedure, particularly in early rounds where the capacity gap between the teacher model and the student model is high. We could certainly replace the teacher model with the smooth labels produced by it for later rounds of B-DISTIL. However, by doing so we lose the effects of non-deterministic preprocessing steps, for instance random rotations and crops for images, on the teacher logits. Distillation is therefore a key component of our approach, as reflected in the title and algorithm name.
>
>
> *[1] Machine learning at the network edge: A survey. Murshed, MG Sarwar, et al. ACM Computing Surveys (2021)*
>
> *[2] Visual Wake Words Dataset. Aakanksha Chowdhery, Pete Warden, Jonathon Shlens, Andrew Howard, Rocky Rhode. Arxiv: 1906.05721 (2019).*
>
> *[3] Dynamic Neural Networks: A Survey. Yizeng Han, Gao Huang, Shiji Song, Le Yang, Honghui Wang, Yulin Wang. IEEE Transactions on Pattern Analysis and Machine Intelligence (2022).*

---

> > ### Author Response · Authors · 2023-08-18
> > **Additional questions**
> >
> > Thank you again for taking the time to review our submission. We hope our responses have resolved the reviewer's concerns but we are happy to discuss further if the reviewer has any additional questions.

---

> > ### Comment · Reviewer_85M5 · 2023-08-21
> >
> > Thank you for your detailed response. I appreciate that you have addressed some of my concerns, and as a result, I am willing to raise my score.

---

### Official Review · Reviewer_2TpN · 2023-07-07

**Soundness:** 3 good
**Presentation:** 1 poor
**Contribution:** 2 fair
**Rating:** 5
**Confidence:** 4

**Summary:**

The authors describe a new method for knowledge distillation with an ensemble of lower capacity student models, and draw connections between their method and classical boosting approaches. They provide a theoretical analysis of the risk of this method, and demonstrate the benefits of their approach in learning tasks with a limited inference budget, either in terms of computing cost or speed of inference.

**Strengths:**

 A compelling idea. Approaching the task of ensemble distillation through a boosting perspective is to my knowledge a novel idea, and the connections that can be made to the boosting literature as a result are quite interesting. At face value, the B-DISTIL algorithm does not seem limited to the distillation setting, and it would be interesting to understand how it compares more generally to other boosting algorithms.

**Weaknesses:**


As an overall weakness, I found the presentation of the paper to be very confusing. In particular:
- The relationship between two-player games and boosting needs to be better described in the related work and problem formulation. Schapire and Freund do not simply "show that weak learners can be aggregated to produce strong learners" (lines 90,91) but rather establish key correspondences between the formulation of two player games and boosting algorithms. Language from this correspondence are used throughout section 3.1 ("players,", "minimax value of the game", "ensemble of predictors","weak learners") without a clear description of how two player games and boosting relate to one another, and make the exposition difficult to follow.
- Notation needs to be more carefully defined throughout the text. For example, the constants $L,M,N,R$ are all used without explicit reference to what they represent in the main text (see questions below for more notation issues).
- It is unclear to me how student models are constructed. Do the configurations in Tables 2-7 provide specifications for the student models used? Or rather for $\mathcal{F}_0$? Are the different rows of the table different base models, or do they somehow relate to the use of intermediate layer connections? How are intermediate connections implemented in each of the specific model architectures described?

Overall, I believe that the quality of the paper suffers significantly from issues with presentation, and I am willing to reconsider my score if these issues are addressed.

Another weakness of this work is comparison to previous work with respect to experimental findings. It would be useful to know how the results in Figure 4 compare to anytime inference as described in Huang et al. 2018 or Ruiz and Verbeek, 2020 for image classification results.

**Questions:**

- What is an (ordered) set of low inference cost hypothesis classes? (Lines 115-117). Does this correspond to an ensemble with an increasing number of member models? If so it should be clearly stated that this is the case. How does $\mathcal{F}$ relate to $\mathcal{F}_m$? (Line 121).
- How does $\mathcal{F}_r$ relate to $\mathcal{F}_0$? How is $\mathcal{F}'_r$ related to $\mathcal{F}_r$?
- On first glance, the weak learning condition (Definition 1) appears to me quite different from other weak learning conditions in the traditional boosting literature. Is there an interpretation of this condition that is consistent with other definitions of weak learning? This would be good to know.
- The claim "Existing boosting methods for classification treat multi-class settings (L > 1) as L in stances of the binary classification problem (one vs. all)" (Lines 162-166) does not apply to Adaboost.M1. Are the weak learners that you study here unable to meet the weak learning condition for Adaboost.M1? If they meet this condition, it would be a useful baseline against which to compare the performance of this method.
- In Figure 3, it would be useful to know the correspondence between the total number of FLOPS required by the ensemble, and the corresponding model accuracy.


**Limitations:**

As the authors state, some limitations of their methods include the need to design the class of student models, and the potential additional cost of evaluating models.

---

> ### Author Rebuttal · Authors · 2023-08-09
>
> Thank you for your many great suggestions for improving the presentation of our work, particularly the problem formulation.
>
> **Boosting, two player games: relationship, terminology and notation.** Due to space limitations we shortened the exposition on boosting, zero-sum games and weak-to-strong learning. However we agree with the reviewer that the relationship between these notions are much deeper and intricate than just “aggregating weak learners to strong learners”. In fact, the connections between online and multi-objective optimization and model compression, as seen for example in dense model theorems and regularity lemmas from complexity theory (see exposition by Luca Trevisan [1] and Theorem 2 in particular) was a key motivation for our work. We will include a detailed discussion on the connections between these notions in the appendix with the aim both to improve exposition and also to encourage the flow of ideas between the communities. We will also include an introduction of these ideas in the main text, casting classical boosting as a two-player game with the aim of introducing the terminology and notation.
>
> **Student models: ordered classes, connections and model configurations.** The configurations specified in Tables 2-7 are of the *unique* student models used for the distillation run. Note that the same student model configuration may be reused in successive rounds. The corresponding hypothesis classes are ordered from low-inference latency to high when moving from top to bottom in these tables. The connections used are implemented by overloading the forward pass of the candidate models at training time [code: `ddist.candgen:op_add_connections`]. The connections themselves are all-reduce-like (stateless) operations [code: `ddistexps.autosearch.gen_fwds`]. $\mathcal{F}_r$ and $\mathcal{F'}_r$ are recursively defined with $r=0$ specified by $\mathcal{F}_0$. We will update the tables to make this clear and include a discussion of how the connections are implemented to the appendix.
>
> **Weak learning condition and AdaBoost.M1.** At a conceptual level, both the notion used in our paper and the standard weak learning guarantees are very similar. Both guarantees require that for an arbitrary reweighting (represented by K in our papers) we are able to find correlations between the hidden function (represented by the labels in the usual boosting set up and by the teacher network in our set up). From this perspective, both algorithms can be seen as “boosting” weak correlations to strong correlations through iterative reweighting. For binary classification, B-DISTIL and AdaBoost can be shown to have the same weak learning condition. For multi-class settings, they have a similar weak learning condition, which strictly speaking is stronger than the weak learning condition used in the binary classification setting (see, Section 5 in [2]). The main difference between the methods is that while AdaBoost.M1 only considers the prediction outputs w.r.t labels, $I(y_i == h(x_i))$, we work with the teacher logits and the difference between the logits $f(x_i) - h(x_i)$.  Adaboost.M1 abstracts away the details of finding a weak learner for a particular data distribution as part of a subroutine WeakLearn [2]. As we note in the main draft, finding weak learners in this sense is difficult [3], but since we use a smooth loss and a weighting tied directly to logits, we enjoy the advantage of being able to employ distillation directly on the residuals and the weak learning condition (Equations 6-7) in FIND-WL. We will add the correspondence between the weak learning conditions into the appendix.
>
> We will also modify Figure 3 to also include actual values; thank you for this suggestion. The required FLOPS values (in million FLOPS) profiled for one vision and time-series data in Figure 3 is provided below.
>
> *CIFAR10*
>
> | Model | Connections |
> |-----------|--------------|
> | 7.37    | 0.0                |
> | 44.96  | 0.14            |
> | 63.2   | 0.71            |
> | 63.2 | 1.53             |
> | 63.2  |2.34.            |
>
> *Google-13*
>
> |Model | Connections |
> |---|--|
> |0.084 | 0.000 |
> |0.655 | 0.013 |
> |1.051 | 0.058 |
> |1.83. | 0.120 |
> |1.82  | 0.211 |
>
> **MSDNets and HNE:** Thanks for bringing up Huang et al. 2018 (MSDNets) and Ruiz and Verbeek, 2020 (HNE), which we cite in our related work. We do not compare to these directly as they can essentially be seen as restricted variants of our approach. By picking the base class as dense networks at various scales as in Figure 2 (Huang et al. 2018), and connections as dense connections our algorithm can recover MSDNets from Huang et al. 2018. Similarly, by picking the base class as root nodes (Figure 1 Ruiz and Verbeek, 2020), and connections as binary connections, we recover an HNE. Our method formalizes this intuition of a graph comprising of a base-class and a specific connection, and provides theoretically motivation within the framework of boosting and two player games for a training procedure to construct such structures. This allows us to generalize these notions, consider a broader set of connections (ex. residual connections) and base classes (ex. recurrent networks) and data modalities,  while taking into account their implementation costs when deciding on the final structure. The two mentioned works are closely tied to image datasets for anytime inference and are not readily applicable for sequential inference. They are also only designed to optimize for latency. Finally, we note that we do compare against the more related approach of E-RNN in our early-prediction experiments.
>
> *[1] Online Optimization:  Regularity Lemmas. Luca Trevisan, In-theory (Online), 2019.*
>
> *[2]: A Decision-Theoretic Generalization of On-Line Learning and an Application to Boosting. Yoav Freund and Robert E. Schapire . Journal of Computer and System Sciences (1996).*
>
> *[3]: Generalized Boosting. Arun Suggala, Bingbin Liu, Pradeep Ravikumar. NeurIPS (2020).*

---

> > ### Comment · Reviewer_2TpN · 2023-08-15
> > **Thank you.**
> >
> > Thank you for your response. I appreciate the detailed answers to the points mentioned, and I believe my concerns about presentation will be addressed by the revisions suggested by the authors in their rebuttal. I especially support the revisions suggested around the expositions of student models. I have updated my score to reflect these revisions.

---

### Official Review · Reviewer_KanN · 2023-07-07

**Soundness:** 3 good
**Presentation:** 3 good
**Contribution:** 3 good
**Rating:** 6
**Confidence:** 2

**Summary:**

The main focus of this paper is to address a problem in progressive knowledge distillation, which involves approximating a single large teacher model by utilizing an ensemble of multiple smaller student models. The authors propose an algorithm called B-DISTILL to tackle this specific problem. One notable advantage of this methodology is its capability to effectively balance the trade-off between cost and performance by adjusting the ensemble size of the student models.

**Strengths:**

1. The problem formulation of "progressive knowledge distillation" is intriguing and well-motivated. In conventional knowledge distillation approaches for model compression, small student models of fixed sizes are typically employed, resulting in a fixed inference cost. A notable advantage of the proposed methodology is its ability to dynamically adjust the inference costs based on the available resources, which is a clear strength of the approach.
2. While the concept of approximating a function using a combination of multiple functions is not novel (as evident from classical boosting methods mentioned by the authors), this paper provides a distinct contribution by connecting these ideas to the field of knowledge distillation.

**Weaknesses:**

1. The scalability of the proposed methodology appears to be somewhat limited. It was anticipated that B-DISTILL would achieve a similar level of performance as the teacher model while utilizing the same inference cost. However, when applied to TinyImageNet and ImageNet datasets, B-DISTILL falls short of meeting this expectation.
2. One important baseline is missing - deep ensembles using the model structure considered in B-DISTILL. Including this baseline would provide a clear motivation for the progressive formulation adopted in B-DISTILL.

**Questions:**

1. The proposed methodology is in line with the principles of slimmable networks (Yu et al., 2019), as it enables users to control the inference cost. Although there are distinctions in the primary categories of each approach, such as pruning and distillation, they share common properties in terms of model compression. Thus, it would be advantageous to include related works, such as slimmable networks, in the main text to ensure readers have a comprehensive understanding of the topic.
2. Could we adapt the inference cost based on the "difficulty" of the input? Given that there might be instances where accurate predictions can be made without the need for additional student models, the idea of limiting the ensemble size based on the difficulty is highly appealing.

__Miscellaneous:__
1. Typo: "B-DSTILL" in Figure 2.
2. Adjust the legend in Figure 3.

**Limitations:**

The authors addressed the limitations.

---

> ### Author Rebuttal · Authors · 2023-08-09
>
> Thank you for your thoughtful feedback and positive assessment of our work.
>
> **Scalability:** Although B-DISTIL takes additional inference time to make accurate predictions for ImageNet, the teacher models in this case have 100+ layers.  Tasks in efficient inference that rely on smaller models for image datasets, or scenarios (e.g., edge inference) involving sensor/audio data streams, are key application areas that stand to benefit from B-DISTIL. However, even at larger scales, where the data distribution during inference is skewed towards 'easier' samples, B-DISTIL can be advantageous in the average case, as the majority of the predictions can be completed quickly (Google-13, Table-1, T=50%). In fact our goal in providing the ImageNet scale experiments was to demonstrate scalability of our implementation, especially since maintaining and updating the weight-matrix of a 1000 class, 1M training image dataset can be non-trivial. We will clarify this in the main text and add a discussion about the techniques we use to manage compute requirements (e.g., streaming weights off-disk asynchronously [code: `ddist.data:DataFlowControl`], performing weight updates in log-space [code: `ddist:ClfPlayer.log_space_update()`], using a shared-memory object store [code: `ddist.dispatch` utilizing the `ray` parallelization library ]) in the appendix.
>
>
> **Baselines and slimmable networks**: Thank you for the reference on slimmable networks. We will include a discussion of this and similar works in the related work section. In particular,  slimmable networks take the approach of adjusting the width at inference time based on on-device resource constraints. While we focus more broadly on network hyperparameters, both approaches do share similar ideas when thinking of them as trading off on-device performance vs. inference cost. Regarding baselines, the current NO-RESCHED baseline does use the same model structure for end-to-end distillation as used by B-DISTIL, without any activation sharing/connections. We will clarify this in the main text. We will also add end-to-end distillation performance using the same network architecture along with connections to the appendix.
>
> **Adapting inference cost based on difficulty of input:** We agree that limiting the ensemble size based on difficulty of the input is an exciting prospect. In fact, the application of our method to early-prediction in sequential inference can be interpreted as classifying based on difficulty of the input (Table 1). However, this is an intuition based argument and we leave exploring if the samples classified early have some notion of 'simplicity' as an interesting direction of future work.

---

> > ### Comment · Reviewer_KanN · 2023-08-17
> >
> > Thank you for the authors' efforts and further insights. I keep my positive assessment.

---

### Decision · Program_Chairs · 2023-09-21

**Decision:**

Accept (poster)

**Comment:**

The paper studies the problem of progressive distillation which requires distilling a large pre-trained model into a series of smaller student models such that one can employ a composition of a subset of student models at the runtime to realize the desired inference cost vs. accuracy trade-off. Towards this, the authors propose B-Distill to progressively learn "weak learners" via distillation. Interestingly, the authors allow a particular student model in the progression to leverage the output of intermediate layers of the earlier models in the progression.

Both theoretical and empirical results validate the utility of the proposed progressive distillation method. B-Distill is evaluated on multiple real-life domains (vision, speech, and sensor data) and multiple model architectures are explored. The author presents results for both any-time prediction and early-exit settings.

All of the reviewers acknowledged the value of the contributions in the submission. The reviewers were able to successfully address the key concerns/questions raised by the reviewers during the rebuttal phase. The reviewers made multiple suggestions to improve the quality of the presentation. The authors have promised to incorporate these suggestions in the revised version of the submission. They are strongly advised to act on these suggestions while preparing the camera-ready version. Some of the key points are as follows:

1) Expand the discussion of prior work to include results on pruning, e.g., slimmable networks.

2) Properly introduce various notations before their usage.

3) Provide the necessary background on the connection between boosting and two-player games, as well as how it precisely connects with the proposed method.

4) Improve the discussion of experimental results and highlight how various natural baselines suggested by the reviewers are already covered by the existing results.